# Multiscale Analysis of Runoff Complexity in the Yanhe Watershed

**DOI:** 10.3390/e24081088

**Published:** 2022-08-07

**Authors:** Xintong Liu, Hongrui Zhao

**Affiliations:** 1Institute of Transportation Engineering and Geomatics, Department of Civil Engineering, Tsinghua University, Beijing 100084, China; 23S Center, Tsinghua University, Beijing 100084, China

**Keywords:** runoff complexity, *RCMSE*, CEEMDAN, Yanhe watershed, multiscale

## Abstract

Runoff complexity is an important indicator reflecting the sustainability of a watershed ecosystem. In order to explore the multiscale characteristics of runoff complexity and analyze its variation and influencing factors in the Yanhe watershed in China during the period 1991–2020, we established a new analysis method for watershed runoff complexity based on the complete ensemble empirical mode decomposition with adaptive noise (CEEMDAN) method for the decomposition of multiscale characteristics and the refined composite multiscale entropy (*RCMSE*) method for the quantification of the system complexity. The results show that runoff and its components all present multiscale complexity characteristics that are different from random signals, and the intermediate frequency modes contribute the most to runoff complexity. The runoff complexity of the Yanhe watershed has decreased gradually since 1991, and 2010 was a turning point of runoff complexity, when it changed from a decline to an increase, indicating that the ecological sustainability of this basin has improved since 2010, which was mainly related to the ecological restoration measures of the Grain for Green Project. This study expands the research perspective for analyzing the variation characteristics of runoff at the multiscale, and provides a reference for the study of watershed ecological sustainability and ecological management.

## 1. Introduction

A watershed is a topographically delineated area formed under the internal forces of the earth and modified by external forces and human activities [1]. Watersheds are hydrological response units, biophysical units, and holistic ecosystems, possessing all the complexities of land surface systems, thereby making them excellent candidates for the practice of earth system science [2,3,4]. The natural flow regime is considered the primary driving force behind the formation of habitats and distribution, diversity, and abundance of biota, and it is extremely important for maintaining and sustaining riverine ecosystem integrity and its biodiversity [5]. Climate change and human activities are the two main driving factors that affect water cycles and the evolution of water resources. Frequent and intense human activities, such as afforestation and deforestation, grassland conversion, urbanization, and dam construction, determine rainfall redistribution and alter surface runoff, infiltration, groundwater recharge, instream flow, and evapotranspiration processes [6,7]. Runoff, the key component of the hydrological cycle, is directly or indirectly influenced by numerous types of positive and negative feedbacks at various scales, such as rainfall, climate change, human activities, and other surface factors [8,9], so that the runoff of a watershed is a complex system that is nonlinear, nonstationary, and uncertain [10,11]. Complexity is an essential and core feature of a hydrological system [12]. In-depth exploration of the inherent complexity of runoff is of theoretical and practical significance for revealing the instability of hydrological cycle dynamic processes and the self-organization ability of watershed systems.

Complex system science provides a valuable reference for complexity research into runoff systems. Entropy methods derived from information theory are simple algorithms with high sensitivity, strong robustness against noise, and no assumption of the statistical characteristics of the data [13,14]. By regarding the watershed system as a living organism, the concept that ‘life feeds on negative entropy’ has a profound impact on the study of natural systems [15]. Entropy methods have been widely used in evaluating the complexity of nonlinear and overall hydrological dynamics [11,16,17]. Sample entropy (SE) [18] quantifies the degree of regularity of a time series by evaluating the appearance of repetitive patterns, and has excellent stability and reliability in detecting the randomness and complexity of runoff [19,20]. Complexity is associated with ‘meaningful structural richness’ [21], but SE essentially comprises the statistical analysis of regularity, without detecting the nonlinear characteristics or quantifying the fractal behaviors of signals [22]. Therefore, an increase in SE is related to an increase in irregularity, but does not necessarily mean an increase in system complexity.

Costa et al. [23] introduced multiscale entropy (MSE) analysis to quantify the complexity of biological systems in cardiology. MSE takes into account the multiple temporal scales by the estimation of SE depending on the coarse-graining time series. MSE is based on the observation that the state of a complex system is far from perfect regularity and complete randomness [21] and reveals the structure of long-range correlation on multiple scales by quantifying the multiscale variability of signals [24]. The complexity is usually related to the ability of life systems to adapt to changing environments, which requires integrated multiscale functions. The results of MSE have been proven to be consistent with the ‘complex loss’ of ill-conditioned systems [21,24,25,26]. Similarly, the runoff structure in hydrology also has its own complexity. The more complex the structure, the stronger the self-regulation and restoration ability, which usually means that the watershed is healthier [17]. In general, the original complexity of a hydrological system is close to the maximum that can occur with a long-term evolution of natural conditions, but it may lose its complexity and become an unhealthy watershed system because of human activities, such as soil and water conservation and construction of water conservancy projects [27,28]. Therefore, MSE is also applicable to the complexity study of a runoff system for the measurement of multiscale characteristics of runoff and the system’s adaptability to the environment.

However, since coarse-graining procedures greatly shorten the length of time series, MSE may produce inaccurate entropy estimates or induce undefined entropy [29]. Composite multiscale entropy (CMSE) [30] was proposed to improve the accuracy of MSE, but it does not resolve undefined entropy. Wu et al. [29] proposed a refined composite multiscale entropy (*RCMSE*) to improve CMSE, which improves the accuracy of entropy estimation and reduces the probability of generating undefined entropy, making it more suitable for the analysis of runoff data with a limited sequence length. 

Due to the interaction between various dynamic mechanisms, runoff time series contain various scales of fluctuations and possess complexity of different time scales. Empirical mode decomposition (EMD) is an adaptive signal decomposition method that was proposed by Huang et al. [31]. It assumes that the data may have many different coexisting modes of oscillations in various scales at the same time, and decompose the original series into these intrinsic modes based on the local characteristic scales of data themselves; these components are called intrinsic mode functions (IMFs). The complete ensemble empirical mode decomposition with adaptive noise (CEEMDAN) [32] is an important improvement on EMD. Compared with most EMD improvement methods, CEEMDAN effectively solves the mode mixing problem and generates complete and noise-free reconstruction. Currently, the CEEMDAN method has been widely used in the signal processing field [33,34,35], but it has insufficient applications in hydrology. Combining *RCMSE* with CEEMDAN, the characteristics of runoff time series can be understood sufficiently at the micro and macro levels.

The Yanhe watershed is located in the middle of the Loess Plateau in China, which is a landscape that has been significantly affected by climate change and anthropogenic activities [36,37]. There is an urgent need to evaluate the ecosystem sustainability of this region. The overall aim of this paper was to propose a new research method for watershed runoff complexity based on *RCMSE* and CEEMDAN. The mechanism of multiscale runoff complexity and the variation and influencing factors of complexity in the Yanhe watershed over the last 30 years were studied to provide references for the implementation of ecological conservation and watershed management.

## 2. Materials and Methods

### 2.1. Study Area

As a primary tributary of the Yellow River, the Yanhe River has a total length of 286.9 km. The Yanhe watershed (36°27′–37°58′ N, 108°41′–110°29′ E) is located in the hinterland of the Loess Plateau, with a total area of 7687 km^2^ and an altitude of 491–1787 m, as illustrated in Figure 1. The Yanhe watershed has a typical loess landform with crisscross ravines, loose soil, and poor antierosion ability. This region is a semiarid continental climate zone, with a mean annual precipitation of about 520 mm and a multiyear mean temperature ranging from 8.8 to 10.2 °C. The seasonal distribution of precipitation is quite uneven; more than 75% occurs between June and September as rainstorms. In the past, due to the influence of unreasonable anthropogenic activities and natural factors, the ecosystem in this region was significantly degraded, with a sharp decrease in natural vegetation and severe soil erosion, resulting in serious impacts on regional sustainable development [36]. Consequently, the Grain for Green Project, which includes a series of ecological construction policies, has been carried out in this region since 1999 [37]. 

### 2.2. Data Sources

The data used in this study included the following: (1) daily runoff data for the period of 1991–2020 were collected from the Ganguyi hydrological station, which is the control hydrographic station and the hydrological calibration outlet in the research basin with a control area of 5891 km^2^, accounting for about 76.6% of the basin area [38,39]. All the runoff data came from the hydrological yearbook of the Yellow River Basin provided by the Yellow River Conservancy Commission of the Ministry of Water Resources [40]; (2) digital elevation model (DEM) data with 30 m resolution, obtained from the Geospatial Data Cloud [41]; and (3) Yan’an Statistical Yearbook data for 2020, obtained from the Yan’an Bureau of Statistics [42].

### 2.3. Methods

#### 2.3.1. Refined Composite Multiscale Entropy

Multiscale entropy is an effective method used to measure the complexity of a time series and has been applied in many fields successfully [21,23,24,25,26], but as the time scale factor increases, MSE may yield an inaccurate estimation of entropy or induce undefined entropy. Composite multiscale entropy (CMSE) [30] was proposed to improve the accuracy of entropy estimation, but CMSE increases the probability of inducing undefined entropy. In 2014, Wu et al. [29] proposed refined composite multiscale entropy (*RCMSE*) to improve MSE and CMSE for the undefined entropy problem of short time series. The *RCMSE* algorithm consists of the following three procedures:

(1) For a discrete time series $x=\left\{x_{1},x_{2},\ldots ,x_{N}\right\}$, after the initial normalization of the original series, consecutive coarse-graining procedures are performed at different scales, and the coarse-grained sequence represents the system dynamics at different time scales. For a scale factor *τ*, the *k*-th coarse-grained time series is defined as $y_{k}^{\left(\tau \right)}=\left\{y_{k,1}^{\left(\tau \right)},\;y_{k,2}^{\left(\tau \right)},\ldots ,y_{k,p}^{\left(\tau \right)}\right\}$, where:(1)$$y_{k,j}^{\left(\tau \right)}=\frac{1}{\tau }\sum_{i=\left(j-1\right)\tau +k}^{j\tau +k-1}x_{i},\;\;\;\;\;\;\;\;\;1\leq j\leq \frac{N}{\tau },1\leq k\leq \tau .$$

(2) For all coarse-grained time series of each scale factor *τ*, the numbers of similar vector pairs $n_{k,\tau }^{m+1}$ and $n_{k,\tau }^{m}$ are computed, where $n_{k,\tau }^{m}$ represents the total number of *m*-dimensional vector pairs from the *k*-th coarse-grained time series for a scale factor *τ* for which the distance between the two vectors is smaller than a predefined tolerance *r* [18]. Referring to the relevant literature [21], for larger m, both the SE and the coefficient of variation increase dramatically due to the finite number of data points, and for larger r, fewer vectors are distinguishable, so we used m=2 and r=0.15σ, where σ denotes the standard deviation (SD) of the original time series, and m and r both remain constant for all scales.

(3) Let ${\bar{n}}_{k,\;\tau }^{m+1}$ and ${\bar{n}}_{k,\;\tau }^{m}$ represent the mean of $n_{k,\tau }^{m+1}$ and $n_{k,\tau }^{m}$, respectively, for 1≤k≤τ. At a scale factor *τ*, the *RCMSE* value is defined as the logarithm of the ratio of ${\bar{n}}_{k,\;\tau }^{m+1}$ to ${\bar{n}}_{k,\;\tau }^{m}$, which is provided as Equation (2):(2)$$RCMSE\left(x,\tau ,m,r\right)=-\ln\left(\frac{\;{\bar{n}}_{k,\;\tau }^{m+1}}{{\bar{n}}_{k,\;\tau }^{m}}\right)$$
where ${\bar{n}}_{k,\;\tau }^{m+1}=\frac{1}{\tau }\sum_{k=1}^{\tau }n_{k,\tau }^{m+1}$ and ${\bar{n}}_{k,\;\tau }^{m}=\frac{1}{\tau }\sum_{k=1}^{\tau }n_{k,\tau }^{m}$. Equation (2) can be simplified as:(3)$$RCMSE\left(x,\tau ,m,r\right)=-\ln\left(\frac{\;\sum_{k=1}^{\tau }n_{k,\tau }^{m+1}}{\sum_{k=1}^{\tau }n_{k,\tau }^{m}}\right)$$

The *RCMSE* is used to compare the relative complexity based on the following guidelines [21]: (1) If for most scales the entropy measures are higher for one time series than for another, the former is considered more complex than the latter; (2) a monotonic decrease in entropy measures indicates that the original signal only contains information at the smaller scales. Therefore, in the analysis of the complexity of the runoff system, not only the specific entropy values but also their dependence on scales needs to be considered, such as the areas under the *RCMSE* curves and the morphological characteristics of *RCMSE* curves.

#### 2.3.2. Complete Ensemble Empirical Mode Decomposition with Adaptive Noise

Huang et al. [31] proposed the empirical mode decomposition (EMD) to decompose the complex time series into intrinsic mode functions (IMFs). The EMD has great advantages in dealing with nonstationary and nonlinear signals, but it still has a ‘mode mixing’ problem, which refers to the presence of similar oscillations in different modes or disparate amplitudes in a mode [43]. Therefore, the ensemble EMD (EEMD) [44] adds the Gaussian white noise to eliminate the mode mixing in the EMD. However, along with introducing the Gaussian white noise, the EEMD algorithm cannot completely eliminate Gaussian white noise after signal reconstruction, and it probably generates a different number of IMFs after adding different noise. Consequently, the complete ensemble empirical mode decomposition with adaptive noise (CEEMDAN) [32] was proposed as an improved version of EEMD. The CEEMDAN adds Gaussian white noise to each stage of the decomposition process, and each IMF is calculated by averaging the results, obtaining decomposed components with less noise and more physical meaning [33,34]. The CEEMDAN process proceeds as follows:

(1) Add Gaussian white noise to the original data $x\left(t\right)$ to create new time series and use the EMD method [31] to obtain the first IMF, ${\mathrm{IMF}}_{1}$, and the first residue, $r_{1}$.

(2) The following *k*-th IMF (k≥2) ${\mathrm{IMF}}_{k}$ and residue $r_{k}$ can be obtained by:(4)$${\mathrm{IMF}}_{k}\;\left(t\right)=\frac{1}{N}\sum_{i=1}^{N}E_{1}\left(r_{k-1}\left(t\right)+\varepsilon_{k-1}E_{k-1}\left(G^{i}\left(t\right)\right)\right)$$
(5)$$r_{k}\left(t\right)=r_{k-1}\left(t\right)-{\mathrm{IMF}}_{k}\left(t\right)$$
where *N* is the number of ensemble members, that is, the number of different realizations of white Gaussian noise; $G^{i}$ is the *i*-th Gaussian white noise to be added; and $\varepsilon_{k-1}$ is the signal-to-noise ratio between the additional noise and original signal. Define the operator $E_{j}\left(\cdot \right)$ that produces the *j*th mode obtained by EMD.

(3) Iterate Step 2 until the obtained residue can no longer be decomposed. The original sequence can be computed as:(6)$$x\left(t\right)=\sum_{k=1}^{K}{\mathrm{IMF}}_{k}\left(t\right)+R\left(t\right)\;$$
where *K* is the total number of IMFs, which comprise the characteristics of the original signal at different time scales, and *R* is the final residue, which clearly shows the trend in the original sequence.

The noise standard deviation was set to 0.2, the number of ensemble members *N* was 100, and the maximum number of sifting iterations was 500 in this paper, which were typically used in practice.

## 3. Results

### 3.1. Multiscale Complexity Characteristics of Runoff

The *RCMSE* under scale factors from 1 to 365 d for the daily runoff data during the period of 1991–2020 for the Ganguyi hydrological station were calculated, as shown in Figure 2. When *τ* < 90 d, the entropy measure gradually increased, with an increase in *τ* until it reached the maximum among all 365 scales at about *τ* = 90 d, and it remained stable when $\tau \in \left[90,\;110\right]\;d$. Then, it decreased rapidly until *τ* = 170 d, and there was a sudden drop near *τ* = 120 d. When $\tau \in \left[170,\;210\right]\;d$, it showed a slight increase; in that period, a minimum point of sudden drop appeared again near *τ* = 180 d. Entropy then decreased gradually after *τ* = 210 d.

To further analyze the runoff at different temporal scales, the CEEMDAN method was applied to the daily runoff data during the period 1991–2020 for the Ganguyi hydrological station, and 14 IMFs and 1 residual term (RES) were obtained, as shown in Figure 3. To find the statistical characteristics of each IMF, the mean periods were calculated, which were derived by dividing the total number of points by the number of peaks (see Table 1). The fluctuation characteristics of all IMFs were different, and as the IMF number increased, both frequency and amplitude reduced. The mean periods of IMF1–IMF4 were lower than 10 d as high-frequency modes, and most of short-term strong runoff was decomposed into these IMFs. The mean periods of IMF5–IMF10 were between 10 d and 1 year as the intermediate-frequency modes. The periods of IMF11–IMF14 were longer than 1 year as the low-frequency modes, representing the influence of long-term factors. The RES presented a pattern of slow change around the long-term average, which shows that the runoff gradually decreased from 1991 to 2009, and the decline was faster after 1994; then it slowly increased after 2009.

The *RCMSE* was calculated for all IMFs, as shown in Figure 4. The scales were from 1 d to 90 d because the entropy measures of the original runoff series reached the maximum around *τ* = 90 d (see Figure 2). Figure 4 shows that the high-frequency modes (IMF1–IMF4) have low entropy values and fluctuations at almost all scales, with low complexity. The entropy values of IMF5–IMF9 gradually increased within a certain range, and then decreased after reaching the maximum. The *RCMSE* curves of IMF10–IMF14 gradually increased under scales 1–90 d due to their large periods and long-range correlations. IMF9 and IMF10 had larger summations of the entropy values over research scales than the others (see Table 2), and maintained a growth trend over a wide range so that they made the greatest contribution to runoff complexity. At small scales, intermediate-frequency and high-frequency modes are the dominant modes of runoff complexity. The contribution of low-frequency modes to runoff complexity increased gradually with increasing temporal scales.

We next tested the hypothesis that due to the complexity of runoff series, they cannot be generated by uncorrelated random processes. The complexity of the original runoff series and its IMFs was compared with that of the randomized time series obtained by shuffling the order of original data points. Because, by construction, both the hydrological and the shuffled time series had the same mean, variance, and distribution, any differences in the complexity indexes were caused by differences in the temporal order of the data points and their correlation properties.

The comparison results for the *RCMSE* curves of the original series with the average curves obtained by randomly shuffling 30 times are shown in Figure 5. The *RCMSE* curves of the original and shuffled series all have large differences in numerical size and trend. The *RCMSE* curves of disordered time series should be expressed as entropy values monotonically decreasing with scale factors [21], just like those of the shuffled series of IMF8–IMF14. However, the *RCMSE* curves of the shuffled series of IMF1–IMF7 showed a rapid increase in the initial stage, and then decreased gradually, like those of uncorrelated noise, resulting in a short ‘fake complexity’ phenomenon. This was mainly due to the seasonal heavy rainfall in the Yanhe watershed, which often leads to an explosion of runoff, and these extreme values were mainly decomposed into the high-frequency modes by the CEEMDAN method, leading to a large fluctuation in these IMFs. The shuffling treatment distributes these extreme values relatively evenly throughout the sequence, resulting in an increase in entropy at small scales, but in the long run, entropy still conforms to the characteristics of uncorrelated noise. Moreover, the shuffled series present smoother *RCMSE* curves without small fluctuations, while those of the original sequences often fluctuate with the scale change, such as IMF10 under scales 60–110 d and IMF5 under scales 1–20 d. The above results show that the original runoff series and IMFs all have unique and high complexity.

### 3.2. Trend Analysis of Runoff Complexity

In order to analyze the temporal evolution of runoff complexity in the Yanhe watershed over the past 30 years, considering that the mean period of the slowest fluctuation, IMF14, is about a decade (see Table 1), the *RCMSE* of the runoff series in sliding windows of 10 years, shifting the window by 1 year, was calculated, and the summations of the entropy values over the scales 1–90 d were taken as complexity, and the fifth year of each time period was set to represent the sequence in Figure 6. This shows that the runoff complexity in the Yanhe watershed presented a downward trend from 1995 to 2010. After 2010, the runoff complexity changed from a decline to an increase.

In order to explore the variation in runoff complexity under different temporal scales, the complexity of each IMF was calculated through *RCMSE* curves in sliding windows of 10 years, shifting the window by 1 year. It should be noted that although the complexity performance at different temporal scales has different characteristics in terms of *RCMSE* curves, when analyzing the variation in each IMF, the summations of the entropy values over the same scale range of 1–90 d can be used as a measure of complexity. To make the results more comparable, we divided the complexity of each sequence by the complexity of the first 10 years, as shown in Figure 7. It can be seen that the complexity of high-frequency modes (IMF1–IMF4) and intermediate-frequency modes (IMF5–IMF10) showed a flat or slightly upward trend before 1997, and gradually decreased from 1997 to 2010. After 2010, the variations in the complexity of high-frequency modes were relatively flat, while those of medium-frequency modes gradually increased, similar to the change trend for the original runoff complexity (see Figure 6). The changes in complexity of the low-frequency modes (IMF11–IMF14) did not show a uniform and obvious varying regularity, and presented little correlation with the variation in runoff complexity. Therefore, it can be concluded that the change in runoff complexity was mainly due to the intermediate-frequency and high-frequency components, and the influence of low-frequency components was slight.

## 4. Discussion

### 4.1. Characteristics of Runoff Complexity

In this paper, the watershed system was considered a complex system, assuming the following: (1) the watershed system has complexity characteristics across multiple temporal scales, (2) the complexity reflects its ability of adaptation and regulation in changing environments, and (3) a ‘sick’ watershed will have reduced adaptability and information carried by output variables. The multiscale entropy method was used to characterize the complexity of the watershed system, and the complexity characteristics of different scale components were studied by empirical mode decomposition. The uncertainty characteristic of runoff had two peak areas within 1 year around the scales of 100 and 210 d, and it reached the maximum at the 90-day scale, which was probably related to the correlation time and the period of possible nonlinear oscillations of runoff series. The runoff was decomposed into 14 IMFs with temporal scales from 3 d to 10 years by CEEMDAN. Each component had separate physical meanings and complexity characteristics that were completely different from random signals. The high-frequency IMFs with short periods and large amplitudes represent short-term fluctuation of runoff, which have the minimum multiscale complexity. The low-frequency IMFs with long periods and small amplitudes signify the components of slow variation, which may be mainly affected by atmospheric circulation or celestial activities. The intermediate-frequency IMFs, especially IMF9 of the half-year scale and IMF10 of the annual scale, made the greatest contribution to runoff complexity. This also indicates that the complexity research based on the *RCMSE* method should refer to not only the specific numerical size of *RCMSE* curves, but also the trend changes and the difference with those of the corresponding random shuffled series.

### 4.2. Factors Impacting Runoff Complexity

Runoff variation in the hydrological system is affected by multiple factors. Climate change and human activities are two main driving factors that affect water cycles and the evolution of runoff in a watershed [17]. Studies have shown that meteorological factors have affected the runoff complexity of the Yellow River Basin, and precipitation has the greatest impact, followed by evaporation and temperature [17,45]. However, the literature and actual meteorological data do not indicate that meteorological factors underwent abrupt changes around 2010, which was the turning point of runoff complexity (see Figure 6). Therefore, meteorological factors may affect runoff complexity to a certain extent, but are not the main factors.

Human activities in the Yanhe watershed mainly include the Grain for Green Project and urbanization. The impacts of the Grain for Green Project on watershed runoff can be explained based on two aspects. On the one hand, since 1999, a series of ecological construction projects have been carried out in this area. Research shows that the Yanhe watershed has had increases in areas of woodland and grassland and significant decreases in the amount of soil and water loss, and the quality and service function of the ecosystem have improved [36]. Following the ecosystem undergoing a growth period, the improvement effect on runoff complexity may have lagged behind, so the runoff complexity changed from a decline to an increase after 2010. On the other hand, the Grain for Green Project had two phases: 2000–2010 and 2010–2020. In the early stage of the project, rapid progress of returning farmland to forest and grassland was carried out with a lack of scientific planning and demonstration, resulting in low vegetation survival rate and damaged plots [46,47], which may lead to further reduction of runoff complexity. Based on in-depth field research and scientific planning, a series of improvement policies of the Grain for Green Project were issued, such as the ‘Notice on Improving the Policy of Grain for Green Project from the State Council’ [48], so that these ecological managements in the latter period may have significantly improved the ecological sustainability of the watershed. The annual afforestation area of Yan’an City is shown in Figure 8, which shows that large-range disorderly afforestation in this area has improved significantly since 2004. Moreover, with the increase in the range of afforestation, runoff ecological status is not necessarily improved. Studies have reported that the growth and development of a large number of artificial vegetations have led to an increase in evapotranspiration and a decrease in surface water resources in the Loess Plateau [49]. The complexity of runoff is a comprehensive variable reflecting the ecological status of the basin, which is expected to be an important reference index for ecological sustainability. In addition, Yan’an City has undergone accelerated urbanization in recent years, which may also have affected the runoff complexity to some extent.

## 5. Conclusions

Complexity has always been the focus and difficulty of watershed system science research, which is closely related to the sustainability of a watershed. Regarding the watershed system as a complex system with multiscale characteristics, in this paper, we established a complexity analysis framework for watershed runoff based on CEEMDAN for the decomposition of multiscale characteristics and *RCMSE* for the quantification of system complexity. The results show that the runoff and each temporal-scale component present completely different complexity characteristics exhibited in the numerical size and trend changes of *RCMSE* curves, which verifies that the runoff sequence has multiscale complexity. The high-frequency components with short periods and large amplitudes represent short-term fluctuation of runoff, which may contribute to understanding the response of runoff to short-term interference, such as rainstorms. The low-frequency components with long periods and small amplitudes signify the components of slow variations, which may be mainly affected by atmospheric circulation or celestial activities. The intermediate-frequency components, especially the components with mean periods of half-year and annual scales, make the greatest contribution to runoff complexity, which are the key components in the study of runoff complexity variation. The runoff complexity of the Yanhe watershed has shown a downward trend since 1991, but with a gradual increase after 2010, indicating that the ecological sustainability of this basin improved after 2010, which was probably related to the ecological restoration measures of the Grain for Green Project, showing that the measures in the past decade have effectively improved the degradation phenomenon of runoff complexity. This study has expanded the research perspective in relation to multiscale runoff complexity and the variation characteristics of runoff systems. It also provides a reference for the evaluation of watershed ecological sustainability and ecological management.

The following issues require attention in the future. In this paper, only runoff data for the Ganguyi hydrological station in the Yanhe watershed were considered; future research should introduce data from more hydrological stations to analyze the spatial multiscale characteristics of runoff complexity. In addition, we only analyzed the causes of runoff complexity change qualitatively; the quantitative contributions of climate change and human activities to runoff changes need to be combined in the future.

## Figures and Tables

**Figure 1 entropy-24-01088-f001:**
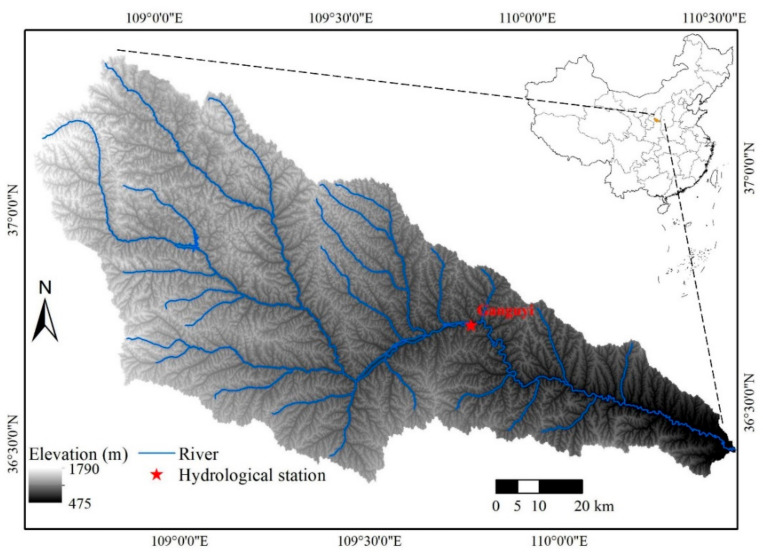
Basic information map of the Yanhe watershed.

**Figure 2 entropy-24-01088-f002:**
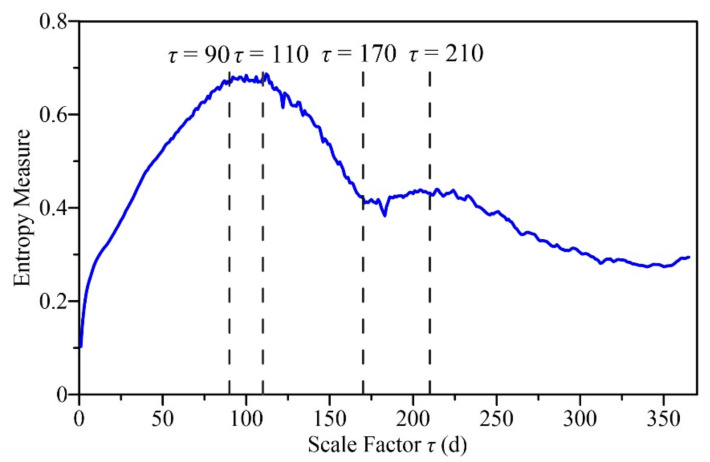
The refined composite multiscale entropy (*RCMSE*) curve for the daily runoff data during the period 1991–2020 for the Ganguyi hydrological station in the Yanhe watershed.

**Figure 3 entropy-24-01088-f003:**
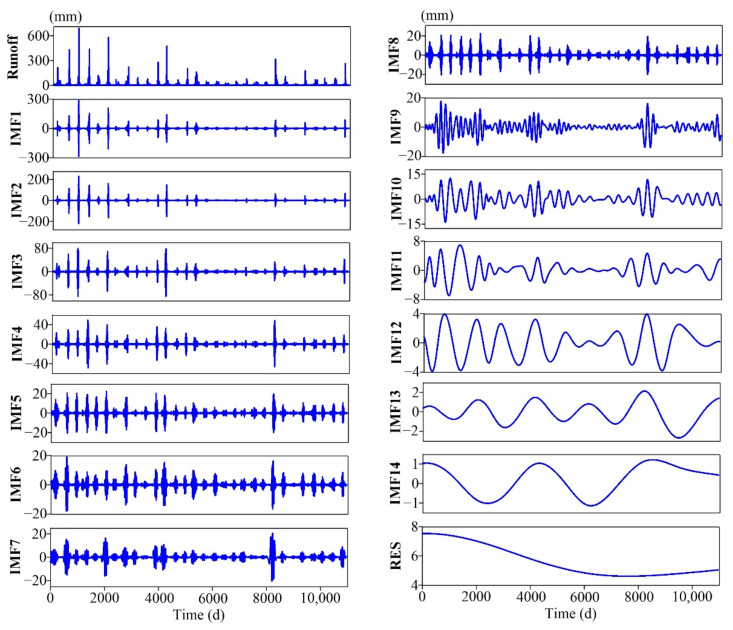
The intrinsic mode functions (IMFs) and residue (RES) for the daily runoff data during the period 1991–2020 for the Ganguyi hydrological station, through complete ensemble empirical mode decomposition with adaptive noise (CEEMDAN).

**Figure 4 entropy-24-01088-f004:**
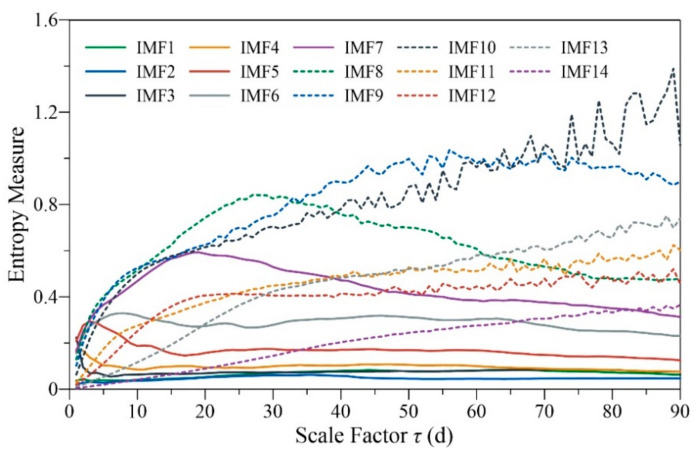
*RCMSE* for IMFs of the daily runoff data during the period 1991–2020 for the Ganguyi hydrological station through CEEMDAN.

**Figure 5 entropy-24-01088-f005:**
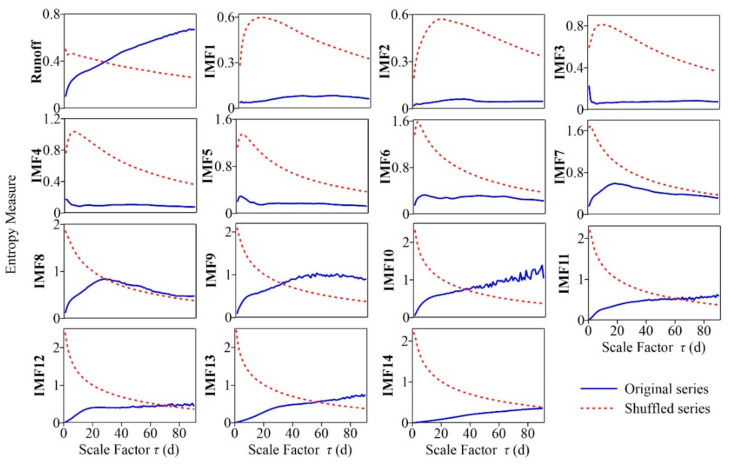
Comparison of the *RCMSE* of the original runoff and IMF series with the corresponding shuffled series.

**Figure 6 entropy-24-01088-f006:**
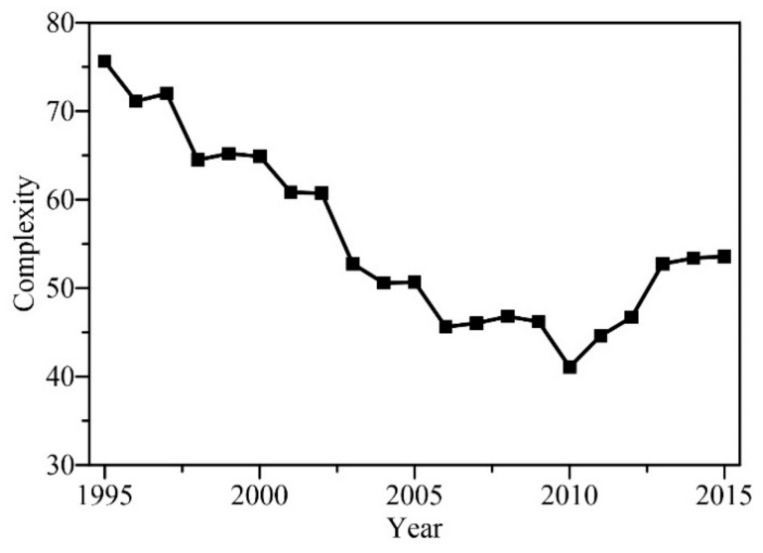
The complexity of runoff in the Yanhe watershed, as shown by sliding windows of 10 years, shifting the window by 1 year.

**Figure 7 entropy-24-01088-f007:**
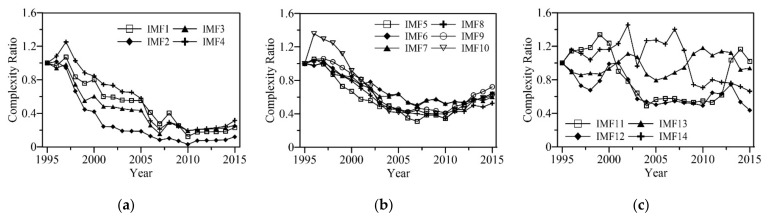
The complexity of IMFs with sliding windows of 10 years and shifting the window by 1 year. (**a**) High-frequency modes (IMF1–IMF4), (**b**) intermediate-frequency modes (IMF5–IMF10), (**c**) low-frequency modes (IMF11–IMF14).

**Figure 8 entropy-24-01088-f008:**
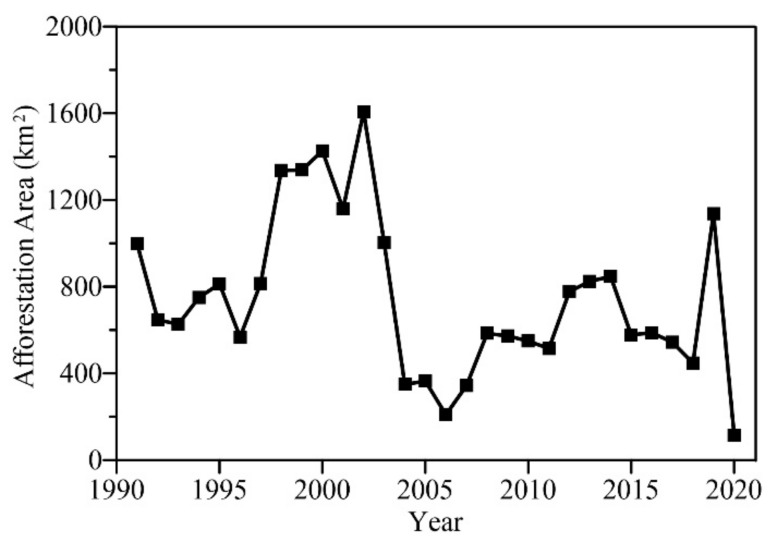
The annual afforestation area of Yan’an City in the period 1991–2020.

**Table 1 entropy-24-01088-t001:** Mean periods of intrinsic mode functions (IMFs) for the daily runoff data from 1991 to 2020 for the Ganguyi hydrological station through CEEMDAN.

	IMF1	IMF2	IMF3	IMF4	IMF5	IMF6	IMF7	IMF8	IMF9	IMF10	IMF11	IMF12	IMF13	IMF14
Mean Period/d	2.91	3.62	3.59	6.45	12.2	22.69	41.67	89.09	171.22	342.44	576.74	1095.80	2191.60	3652.67

**Table 2 entropy-24-01088-t002:** The summations of the entropy values over the scales 1–90 d of *RCMSE* ($RCMSE_{\sum }$) of IMFs of the daily runoff data during the period 1991–2020 for the Ganguyi hydrological station through CEEMDAN.

	IMF1	IMF2	IMF3	IMF4	IMF5	IMF6	IMF7	IMF8	IMF9	IMF10	IMF11	IMF12	IMF13	IMF14
$RCMSE_{\sum }$	6.09	4.23	6.96	8.76	15.18	25.61	38.82	55.38	73.10	72.91	40.67	35.66	40.96	18.40

## Data Availability

Runoff data come from the hydrological yearbook of the Yellow River Basin from the Yellow River Conservancy Commission of the Ministry of Water Resources (http://yrcc.gov.cn/ (accessed on 12 December 2021)). Digital elevation model (DEM) data are openly available in the Geospatial Data Cloud (http://www.gscloud.cn/ (accessed on 6 December 2021)). The Yan’an Statistical Yearbook of 2020 comes from the Yan’an Bureau of Statistics (http://tjj.yanan.gov.cn/Item/5085.aspx/ (accessed on 6 December 2021)) [42]. The datasets generated during this study are available from the corresponding author on reasonable request.

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
