# Peer review of "Multiscale Analysis of Runoff Complexity in the Yanhe Watershed"

_entropy, 2022, doi:10.3390/e24081088_

Round 1
Reviewer 1 Report
Entropy-1800107
Title: Multiscale Analysis of Runoff Complexity in the Yanhe Watershed
Authors: Xintong Liu , Hongrui Zhao
Main comments:
The paper was well written as a technical paper in the data science field, not in the hydrological field. The main concern to publish the paper are:
1) Using one site to represent the hydrology of a watershed is very questionable for the hydrological analysis, not to mention the station is not at the watershed outlet.
2) Typically, daily rainfall is common measured directly in the field, not daily runoff. Please descript how the measurement conduct. Explain the contribute area for the station
3) More explanation about how and why the runoff is the indicator for the sustainability for this watershed, i.e. land cover changes, restoration projects, etc.
Author Response
[General Response]
Thank you very much for your careful review and valuable suggestions on our manuscript, we are quite grateful for that. We have carefully addressed your insightful comments and suggestions, please refer to the new version of this manuscript. The detailed point-by-point responses to the comments are provided below. Please see the attachment for the revised manuscript, and all revisions have been marked up using the “Track Changes” function.
Comments from Reviewer:
Point 1: Using one site to represent the hydrology of a watershed is very questionable for the hydrological analysis, not to mention the station is not at the watershed outlet.
Response 1: Thank you very much for your insightful advice. Ganguyi hydrographic station is a control hydrographic station and the hydrological calibration outlet in the Yanhe watershed with a control area of 5891 km2, accounting for about 76.6% of the basin area [1-4]. Many studies have taken the Ganguyi hydrographic station as the most representative station to study the runoff variation in the Yanhe watershed [2-4]. Therefore, in the case of limited data resources, Ganguyi hydrographic station can be considered to have the ability to reflect complexity of runoff system in Yanhe River Basin. We have added more explanations of the Ganguyi hydrographic station in section 2.2 Data sources of the manuscript. In addition, we agree with you very much that one site may not adequately represent the hydrological situation of the whole basin, but due to the difficulty of obtaining the long-term continuous runoff data of other hydrographic stations in the Yanhe watershed, we only considered the runoff data of Ganguyi station at present. The main focus of this paper is to propose an analysis method of watershed runoff complexity based on multiscale entropy and empirical mode decomposition, and the future research will further consider more hydrological stations and more watersheds for experiments to analyze the spatial characteristics of runoff complexity and provide more comprehensive suggestions to deal with the problem of watershed sustainable development, please refer to the second paragraph of the section 5 Conclusions in the new version of this manuscript.
References
[1] Zhou, Z. X.; Li J. The correlation analysis on the landscape pattern index and hydrological processes in the Yanhe watershed, China. J. Hydrol. 2015, 524, 417–426.
[2] Li, E.H.; Mu, X.M.; Zhao, G.J.; Gao, P.; Shao, H.B. Variation of runoff and precipitation in the hekou-longmen region of the yellow river based on elasticity analysis. Sci. World J. 2014, 2014, 929858.
[3] Lian, Y.; Sun, M.; Wang, J.; Luan, Q.; Jiao, M.; Zhao, X.; Gao, X. Quantitative impacts of climate change and human activities on the runoff evolution process in the Yanhe River Basin. Phys. Chem. Earth. 2021, 122, 102998.
[4] Zhong, X.; Jiang, X.; Li, L.; Xu, J.; Xu, H. The impact of socio-economic factors on sediment load: A case study of the Yanhe River Watershed. Sustainability 2020, 12(6), 2457.
Comments from Reviewer:
Point 2: Typically, daily rainfall is common measured directly in the field, not daily runoff. Please descript how the measurement conduct. Explain the contribute area for the station.
Response 2: Thank you for your helpful advice. All the runoff data come from the hydrological yearbook of Yellow River Basin provided by the Yellow River Conservancy Commission of the Ministry of Water Resources (http://www.yrcc.gov.cn/). The daily runoff from May to October was calculated by the temporary curve method, and the average daily runoff of the fixed flow measurement day was used for the rest of the year. Ganguyi hydrological station is the outlet control station of Yanhe Watershed with a control area of 5891 km2, accounting for about 76.6% of the basin area. We have added more explanations in section 2.2 Data sources in the new version of this manuscript.
Comments from Reviewer:
Point 3: More explanation about how and why the runoff is the indicator for the sustainability for this watershed, i.e. land cover changes, restoration projects, etc.
Response 3: Thanks very much for your valuable advice. The natural flow regime is considered the primary driving forces behind the formation of habitats, and distribution, diversity, and abundance of biota, and is extremely important for maintaining and sustaining riverine ecosystem integrity and its biodiversity [5]. Climate change and human activities are the two main driving factors that affect water cycles and the evolution of water resources. Frequent and intense human activities, such as afforestation and deforestation, grassland conversion, urbanization and dam construction, determines rainfall redistribution and alters surface runoff, infiltration, groundwater recharge, instream flow, and evapotranspiration processes [6,7]. Runoff is the key component of the hydrological cycle, and as a complex system, it is directly or indirectly influenced by numerous types of positive and negative feedback at various scales. Complexity is an essential and core feature of a hydrological system. In-depth exploration of the inherent complexity of runoff is of theoretical and practical significance for revealing the instability of hydrological cycle dynamic processes and the self-organization ability of watershed systems. We have added more explanations about how and why the runoff is the indicator for the sustainability for this watershed, please see the paragraph 1 in section 1. Introduction in the new version.
References
[5] Singh, R.K.; Jain, M.K. Complexity analyses of Godavari and Krishna river streamflow using the concept of entropy. Acta Geophys. 2021, 69, 2325–2338.
[6] Wu, J.; Miao, C.; Zhang, X.; Yang, T.; Duan, Q. Detecting the quantitative hydrological response to changes in climate and human activities. Sci. Total Environ. 2017, 586, 328–337.
[7] Kang, Y.; Gao, J.; Shao, H.; Zhang, Y. Quantitative analysis of hydrological responses to climate variability and land-use change in the hilly-gully region of the Loess Plateau, China. Water 2019, 12, 82.

Reviewer 2 Report
The following comments are given as a reference for revision of the present MS.
1. Page 4: The scale factor τ in Eq. (1) seems to be constant, but it, in Fig. 2, is variable. It is unclear to me. Moreover, what is its physical meaning?
2. Page 4 Lines 41-42: How to calculate the tolerance r? or give the mathematical expression of it.
3. Explain the negative sign on the right-hand side of Eq. (2).
4. Does x in Eq. (5) mean xi or x? Explain it.
5. Line 74: “the number of realizations was 100” is unclear. Explain it.
6. Lines 94-95: The definition of “mean period” should be dimensionless because of “number divided by number”. Nevertheless, the unit of "period" is d in Table. Of course, everyone knows that the unit of "period" is “time”, but according to your definition on Lines 94-95, it is dimensionless.
7. Page 6 Lines 20-21: I cannot figure out from Fig. 3 why the integral of IMF9 and IMF10 was larger. Please list all the integrals from IMF1 to IMF10.
8. The discussion about Fig. 6 is not enough. The authors need to explain more about why 2010 was a turning point. The Grain for Green Project was performed from 2000 to 2020.
9. Page 11 Line 64: In the Conclusion section, the authors reported that “This study has expanded the research perspective … to mechanisms of runoff complexity …”, but I didn’t see anything about “mechanism” in this study.
10. The authors need to strengthen the contribution and new findings in the conclusions section.
Minor comments:
1. Line 15: “for the quantify” is erroneous in grammar.
2. Line 37: The subscript p disappears in Eq. (1). Why?
3. Line 64: Give the definition of IMF when it appears for the first time.
4. Define E1 in Eq. (3). Is E1 the same as E1(Gi)?
5. The unit of Scale Factor is missing.
6. Page 9 Line 99: There may be erroneous grammar in “…to adapt and function in…”.
7. Page 9 Line 16: There may be erroneous grammar in “…with the random shuffled.”
Author Response
[General Response]
Thank you very much for your careful review and valuable suggestions on our manuscript, we are quite grateful for that. We have carefully addressed your insightful comments and suggestions, please refer to the new version of this manuscript. The detailed point-by-point responses to the comments are provided below. Please see the attachment for the revised manuscript, and all revisions have been marked up using the “Track Changes” function.
Comments from Reviewer:
Point 1: Page 4: The scale factor τ in Eq. (1) seems to be constant, but it, in Fig. 2, is variable. It is unclear to me. Moreover, what is its physical meaning?
Response 1: Thanks for your helpful questions. Multiscale entropy (MSE) analysis calculates the entropy measure (Sample entropy) for each coarse-grained time series plotted as a function of the scale factor τ, so that scale factor τ of the refined composite multiscale entropy (RCMSE) method is also variable. The schematic illustration of the coarse-graining procedure is shown in Fig.1 below, and there are two and three coarse-grained time series divided from the original time series for scale factors of 2 and 3 respectively. Given a one-dimensional discrete time series, x={x1,x1,...,xN}, consecutive coarse-graining procedures are performed by different time scales. For a scale factor τ, the consecutive coarse-grained time series, yk(τ)={yk,1(τ),yk,2(τ),...,yk,p(τ)}, are obtained according to Eq. (1) in the manuscript. For scale one, y(1) is simply the original time series. The length of each coarse-grained time series is equal to the length of the original time series divided by the scale factor τ. The coarse-grained sequence represents the system dynamics at different time scales. We have revised this part to make it clearer, please see the second paragraph in Section 2.3.1. Refined composite multiscale entropy in the new version of this manuscript.
Fig. 1. Schematic illustration of the coarse-graining procedure. [1]
References
[1] Wu, S.D.; Wu, C.W.; Lin, S.G.; Lee, K.Y.; Peng, C.K. Analysis of complex time series using refined composite multiscale entropy. Phys. Lett. A 2014, 378, 1369–1374.
Comments from Reviewer:
Point 2: Page 4 Lines 41-42: How to calculate the tolerance r? or give the mathematical expression of it.
Response 2: Thank you for your helpful advice. r=0.15σ, where σ denotes the standard deviation (SD) of the original time series. We have added more explanations for the calculation of the tolerance r in the third paragraph in Section 2.3.1. Refined composite multiscale entropy in the new version of this manuscript.
Comments from Reviewer:
Point 3: Explain the negative sign on the right-hand side of Eq. (2).
Response 3: Thank you for your kind advice. The refined composite multiscale entropy (RCMSE) method is based on sample entropy [2], which is equal to the negative average natural logarithm of the conditional probability that two sequences that are similar for m points remain similar, that is, within a tolerance r, at the next point. For the RCMSE, at a scale factor τ, considering all coarse-grained time series corresponding to different starting points of the coarse-graining process, is equal to the negative average natural logarithm of the conditional probability that the mean of the total number of m+1 dimensional matched vector pairs for scale factor τ divides by that of the m dimensional. We have added more explanations for the calculation of the RCMSE, please refer to the fourth paragraph in Section 2.3.1. Refined composite multiscale entropy in the new version of this manuscript.
References
[2] Richman, J.S.; Moorman J.R. Physiological time-series analysis using approximate entropy and sample entropy. Am. J. Physiol.-Heart Circul. Physiol. 2000, 278, 2039–2049.
Comments from Reviewer:
Point 4: Does x in Eq. (5) mean xi or x? Explain it.
Response 4: Thanks for your helpful questions. The x in Eq. (5) in the old version of the manuscript means the original time series. To make it clearer, we have revised Eq. (6) in the new version of the manuscript (Eq. (5) in the old version of the manuscript) to x(t)=∑k=1KIMFk(t)+R(t) and also modified other parts of the description of CEEMDAN, including the Eq. (4), Eq. (5) and the text, please see the paragraph 2 to 4 in Section 2.3.2. Complete ensemble empirical mode decomposition with adaptive noise in the new version of this manuscript.
Comments from Reviewer:
Point 5: Line 74: “the number of realizations was 100” is unclear. Explain it.
Response 5: Thank you for your helpful advice. The number of realizations means the ensemble member N, that is, the number of different realizations of white Gaussian noise. We have revised this part to make it clearer, please see the paragraph behind Eq. (5) and the last paragraph in Section 2.3.2. Complete ensemble empirical mode decomposition with adaptive noise in the new version of this manuscript.
Comments from Reviewer:
Point 6: Lines 94-95: The definition of “mean period” should be dimensionless because of “number divided by number”. Nevertheless, the unit of "period" is d in Table. Of course, everyone knows that the unit of "period" is “time”, but according to your definition on Lines 94-95, it is dimensionless
Response 6: Thank you for your kind advice. We have deleted the unit from Table 1. Please refer to Table 1 in the new version of this manuscript.
Comments from Reviewer:
Point 7: Page 6 Lines 20-21: I cannot figure out from Fig. 3 why the integral of IMF9 and IMF10 was larger. Please list all the integrals from IMF1 to IMF10.
Response 7: Thank you so much for your helpful comment. Do you mean the integral of IMF9 and IMF10 in Fig. 4 in the manuscript? In this section, we tried to figure out the complexity of runoff at different time scales. Fig. 4 shows the RCMSE curves of IMFs, and IMF9 and IMF10 had larger integrals within the research scales than the others. It may be inferred that IMF9 and IMF10 with the mean periods of 171.22 d and 342.44 d, which can be explained as half-year and annual scales respectively, have the highest complexity, so they may contribute the most to the ability to of adaptation and regulation in changing environments of runoff. All the integrals from IMF1 to IMF14 are listed in the table below, and we have shown them in Table 2 in the new version of this manuscript. In future research, we will further explore why the RCMSE curves of IMFs at various scales differed so much and what specific meanings this difference reflected.
Table 1. The integrals under the 1–90 scales of RCMSE of IMFs of the daily runoff data during the period 1991–2020 for Ganguyi hydrological station through CEEMDAN.
|
|
IMF1 |
IMF2 |
IMF3 |
IMF4 |
IMF5 |
IMF6 |
IMF7 |
IMF8 |
IMF9 |
IMF10 |
IMF11 |
IMF12 |
IMF13 |
IMF14 |
|
Integral |
6.09 |
4.23 |
6.96 |
8.76 |
15.18 |
25.61 |
38.82 |
55.38 |
73.10 |
72.91 |
40.67 |
35.66 |
40.96 |
18.40 |
Comments from Reviewer:
Point 8: The discussion about Fig. 6 is not enough. The authors need to explain more about why 2010 was a turning point. The Grain for Green Project was performed from 2000 to 2020.
Response 8: Thanks so much for your advice. We have discussed the impacts of the Grain for Green Project on watershed runoff in the discussion section from two perspectives. On the one hand, since 1999, a series of ecological construction projects have been carried out in this area. Research shows that the Yanhe watershed has had increases in areas of woodland and grassland and significant decreases in the amount of soil and water loss, and the quality and service function of the ecosystem have improved. Following the ecosystem undergoing a growth period, the improvement effect on runoff complexity may have lagged behind, so the runoff complexity changed from a decline to an increase after 2010. On the other hand, the Grain for Green Project has had two phases: 2000–2010 and 2010–2020. In the early stage of the project, rapid progress of returning farmland to forest and grassland was carried out with a lack of scientific planning and demonstration, resulting in low vegetation survival rate and damaged plots [3,4], which may lead to further reduction of runoff complexity. Based on in-depth field research and scientific planning, a series of improvement policies of the Grain for Green Project was issued, so that these ecological managements in the later period may have significantly improved the ecological sustainability of the watershed. Moreover, with the increase in the range of afforestation, runoff ecological status is not necessarily improved. Studies have reported that the growth and development of a large number of artificial vegetation have led to an increase in evapotranspiration and a decrease in surface water resources in the Loess Plateau [5]. The complexity of runoff is a comprehensive variable reflecting the ecological status of the basin, which is expected to be an important reference index for ecological sustainability. In addition, Yan'an City has undergone accelerated urbanization in recent years, which may also have affected the runoff complexity to some extent. We have added more explanations about why 2010 was a turning point and its relationship with the Grain for Green Project, please refer to the second paragraph in Section 4.2 Factors impacting runoff complexity in the new version of this manuscript.
References
[3] Cao, S.; Chen, L.; Yu, X. Impact of China's Grain for Green Project on the landscape of vulnerable arid and semi‐arid agricultural regions: A case study in northern Shaanxi Province. J. Appl. Ecol. 2009, 46, 536–543.
[4] Xu, X.; Zhang, D.; Zhang, Y.; Yao, S.; Zhang, J. Evaluating the vegetation restoration potential achievement of ecological projects: a case study of Yan’an, China. Land Use Pol. 2020, 90, 104293.
[5] Feng, X.; Fu, B.; Piao, S.; Wang, S.; Ciais, P.; Zeng, Z.; Lü, Y.; Zeng, Y.; Li, Y.; Jiang, X; Wu, B. Revegetation in China’s Loess Plateau is approaching sustainable water resource limits. Nat. Clim. Chang. 2016, 6, 1019–1022.
Comments from Reviewer:
Point 9: Page 11 Line 64: In the Conclusion section, the authors reported that “This study has expanded the research perspective … to mechanisms of runoff complexity …”, but I didn’t see anything about “mechanism” in this study.
Response 9: Thanks so much for your comments. In this paper, we used the multiscale entropy method to analyze the complexity of runoff. From the RCMSE curve of runoff series in Fig. 2 in the manuscript, we can explore the variation of irregularity of runoff sequence with time scales. Combined with CEEMDAN, we analyzed the complexity of runoff components at different time scales, as shown in Fig. 4 and Table 2 in the manuscript. It may not be very appropriate to use the word “mechanism”, so we replaced the sentence “This study has expanded the research perspective in relation to mechanisms of runoff complexity …” with “This study has expanded the research perspective in relation to multi-scale runoff complexity …”, please refer to the reciprocal second sentence of the first paragraph in Section 5. Conclusions in the new version of this manuscript.
Comments from Reviewer:
Point 10: The authors need to strengthen the contribution and new findings in the conclusions section.
Response 10: Thanks for your helpful advice. Complexity has always been the focus and difficulty of watershed system science research, which is closely related to the sustainability of watershed. Regarding the watershed system as a vital organism with multiscale characteristics and self-organizing ability, in this paper, we established a complexity analysis framework for watershed runoff based on CEEMDAN for the decomposition of multiscale characteristics and RCMSE for the quantification of system complexity. The results show that the runoff and each temporal-scale component present completely different complexity characteristics from the random signals exhibited in the numerical size and trend changes of RCMSE curves, which verifies that the runoff sequence has multiscale complexity. The high-frequency components with short periods and large amplitudes represent short-term fluctuation of runoff, which may contribute to understanding the response of runoff to short-term interference such as rainstorms. The low-frequency components with long periods and small amplitudes signify the slowing of varying components, which may be mainly affected by atmospheric circulation or celestial activities. The intermediate-frequency components, especially the components with mean periods of half-year and annual scales, make the greatest contribution to runoff complexity, which are the key components in the research of runoff complexity variation. This study has expanded the research perspective in relation to multi-scale runoff complexity and the variation characteristics of runoff systems. It also provides a reference for the evaluation of watershed ecological sustainability and ecological management. We have strengthened the contribution and new findings in the conclusions section, please refer to the first paragraph in Section 5 Conclusions in the new version of this manuscript.
Comments from Reviewer:
Point 11: Line 15: “for the quantify” is erroneous in grammar.
Response 11: Thanks very much for your careful review and detailed suggestions. We have replaced the sentence “for the quantify the system complexity” with the sentence “for the quantification of the system complexity”, please refer to line 15 in the new version of this manuscript.
Comments from Reviewer:
Point 12: Line 37: The subscript p disappears in Eq. (1). Why?
Response 12: Thanks for your careful question. As the schematic illustration of the coarse-graining procedure is shown in Fig. 1 in Response 1 above, with the increase of k in Eq. (1), the coarse-graining process gradually shifts to the right, and the value of p in Eq. (1) may also change. For example, if N=600 and τ=3, when k=1, p=N/τ=200, but when k=1, p=(N-1)//τ=199. Therefore, based on the above considerations, p was not used in Eq. (1) in the manuscript, and this expression is also consistent with the content of the original literature of RCMSE method [6].
References
[6] Wu, S.D.; Wu, C.W.; Lin, S.G.; Lee, K.Y.; Peng, C.K. Analysis of complex time series using refined composite multiscale entropy. Phys. Lett. A 2014, 378, 1369–1374.
Comments from Reviewer:
Point 13: Line 64: Give the definition of IMF when it appears for the first time.
Response 13: Thanks for your helpful advice. The complete ensemble empirical mode decomposition with adaptive noise (CEEMDAN) is an important improvement on EMD. EMD assumes that the data may have many different coexisting modes of oscillations in various scales at the same time, and decomposes the original series into these intrinsic modes based on the local characteristic scale of data itself; these components are called intrinsic mode function (IMF). We have given the definition of IMF when it appears for the first time, please see the fifth paragraph of Section 1. Introduction.
Comments from Reviewer:
Point 14: Define E1 in Eq. (3). Is E1 the same as E1(Gi)?
Response 14: Thanks for your helpful advice. In Eq. (3) in the old version of the manuscript (Eq. (4) in the new version of the manuscript), the operator Ej(·) means that produces the jth mode obtained by EMD, and we have redefined the Ej(·) to make readers understand the E1(·) and Ek-1(Gi(t) more clearly, please refer to the paragraph below Eq. (5) in the new version of this manuscript.
Comments from Reviewer:
Point 15: The unit of Scale Factor is missing
Response 15: Thanks so much for your advice. We have added the unit of Scale Factor (d) in the corresponding text and the Figure 2, Figure 4 and Figure 5 in the new version of this manuscript.
Comments from Reviewer:
Point 16: Page 9 Line 99: There may be erroneous grammar in “…to adapt and function in…”.
Response 16: Thanks very much for your careful review and detailed suggestions. We have replaced the sentence “the complexity reflects its ability to adapt and function in changing environments” with “the complexity reflects its ability of adaptation and regulation in changing environments”, please refer to the Section 4.1. Characteristics of runoff complexity in the new version of this manuscript.
Comments from Reviewer:
Point 17: Page 9 Line 16: There may be erroneous grammar in “…with the random shuffled.”
Response 17: Thanks very much for your careful review and detailed suggestions. We have replaced the sentence “… the complexity research based on the RCMSE method should not only refer to the specific numerical size of RCMSE curves, but also the trend change and the difference with the random shuffled.” with “… the complexity research based on the RCMSE method should not only refer to the specific numerical size of RCMSE curves, but also the trend change and the difference with those of the corresponding random shuffled series.”, please refer to the last sentence in Section 4.1. Characteristics of runoff complexity in the new version of this manuscript.

Round 2
Reviewer 2 Report
No more comments.
Author Response
Thank you very much for your careful review and valuable suggestions on our manuscript, we are quite grateful for that. We have carefully revised about the background, methods and conclusions in the manuscript, please refer to the new version of this manuscript.
